# Preparation and Properties of Toluene-Diisocyanate-Trimer-Modified Epoxy Resin

**DOI:** 10.3390/polym11030416

**Published:** 2019-03-04

**Authors:** Xiongfei Zhang, Lu Qiao, Xiaolian Lu, Linqi Jiang, Ting Cao

**Affiliations:** Institute of Chemical and Food Engineering, Changsha University of Science and Technology, Changsha 410114, China; QL204551@163.com (L.Q.); m18229978356@163.com (X.L.); jianglinqi0105@163.com (L.J.); ZYF5222012@163.com (T.C.)

**Keywords:** modified epoxy resin, graft copolymerization, 2,4-toluene diisocyanate trimer

## Abstract

In this paper, a novel modified epoxy resin with an interpenetrating network structure for use as a grouting material with high toughness was prepared by a method of graft copolymerization between polyurethane prepolymer (PUP) trimer and epoxy resin (E-44). Polyurethane prepolymer was synthesized using poly(propylene glycol) (PPG) and 2,4-toluene diisocyanate trimer (TDIT) at 70 °C for 3 h. The graft copolymer was prepared by grafting polyurethane prepolymer onto the side chain of epoxy resin at 110 °C. The mechanical properties, fracture surface morphology, chemical structure, thermal properties, and corrosion resistance of the modified epoxy resin curing products were studied. Due to the beneficial flexible segments and the interpenetrating network structure, the results show that when the ratio of epoxy resin to polyurethane prepolymer is 10:2, the optimum mechanical properties are obtained; these include a compressive resistance of 184.8 MPa, impact property of 76.6 kJ/m^2^, and elongation at break of 31.5%. At the same time, the modified epoxy resin curing product also has excellent heat and corrosion resistance. This work provides a new method for the study of epoxy resins with high performance.

## 1. Introduction

Epoxy resin (EP) is a kind of polymer which contains two or more epoxy groups in its molecular chain and forms thermosetting products through various reactions between epoxy groups. It has the advantages of high mechanical strength [1], high bonding strength [2], low shrinkage [3], high stability [4,5], and easy deployment and construction. However, epoxy resin has the disadvantages of poor toughness, brittle quality, easy cracking, and low impact strength, which results in limitations to its application to repairing concrete cracks. Therefore, it is necessary to modify epoxy resin to improve its toughness. The most common toughening method is to toughen epoxy resin with second phase particles, which include rubber elastomer [6], rigid particles [7], nano-particles [8], core–shell polymers [9,10], and thermoplastic resins [11,12]. However, it is difficult to uniformly disperse the additives in the epoxy resin system, which makes the compatibility between the systems poor, leading to the deterioration of the mechanical properties of the materials. Thus, the reaction of polyurethane prepolymer with epoxy resin to form an interpenetrating network polymer has emerged as a toughening method [13,14,15,16,17]. Polyurethane has good compatibility with other kinds of polymers, and these can react independently with each other. Therefore, polyurethane-modified epoxy resin has an obvious toughening effect [18,19,20,21,22,23]. There are still shortcomings in the existing methods. The polyurethane prepolymer is a straight chain, which increases the toughness of the epoxy resin but causes the loss of some of the mechanical properties of epoxy resin.

2,4-toluene diisocyanate trimer (TDIT) is a kind of polymer formed by self-polymerization of 2,4-toluene diisocyanate [24,25,26]. There are six-member ring groups with high rigidity and isocyanate groups with high activity in the molecules of 2,4-toluene diisocyanate trimer [27,28,29]. The highly active isocyanate group has the benefit of easy grafting onto the EP chain, and the special structure of the trimer has the benefit of easily forming a network structure with a high degree of cross-linking. In this paper, with the aim of improving the performance of epoxy resin, a modified epoxy resin with an interpenetrating network structure is prepared by a method of graft copolymerization with 2,4-toluene diisocyanate trimer; we wish to improve the toughness of epoxy resin and, at the same time, maintain or improve its strength.

## 2. Materials and Methods

### 2.1. Materials

Ether of bisphenol A type EP (E-44) and low-molecular-weight polyamide were supplied by Baling Petrochemical Co., Ltd. (Hunan, China). TDIT and Allyl glycidyl ethers (AGE) were purchased from Jining Huakai Resin Co., Ltd. (Jining, China). Polypropylene glycol (PPG 1000) was purchased from Qingdao Chemical Science and Technology Co., Ltd. (Qingdao, China). 2,4,6-tris (dimethylaminomethyl) phenol (DMP-30) and 1,4-butanediol were provided by Chinese Medicine Group Chemical Reagent Co., Ltd. (Beijing, China).

### 2.2. Experiment Procedure

#### 2.2.1. Preparation of the Prepolymer (PUP)

TDIT was placed into a three-necked flask at 40 °C. Then, PPG 1000 was added into the flask, and the mixture was reacted for 3 h at 70 °C. The chain extender was then added and stirring was continued.

#### 2.2.2. Preparation of Modified Epoxy Resin (EP-PUP)

E-44 and polyurethane prepolymer were mixed in a three-necked flask in a certain ratio (given in Table 1), and uniform stirring was maintained at 110 °C. Two hours later, the system was cooled to room temperature, and a transparent liquid could be seen in the flask.

#### 2.2.3. Preparation of the Cured Sample

At room temperature, the curing agent (low-molecular-weight polyamide) was mixed with the modified epoxy resin in a stoichiometric molar ratio of 7:10. Subsequently, DMP-30 (0.2% of the total formulation) was added and stirred vigorously until a homogeneous system was obtained; the mixture was defoamed for 10 min in a vacuum box and then cured naturally.

### 2.3. Characterization Methods

#### 2.3.1. FT-IR Absorption

The modified epoxy was analyzed by Fourier transform infrared spectroscopy (FT-IR) (Perkin Elmer Spectrum 100 FT-IR spectrometer, Waltham, MA, USA) using air as the background and scanning from 4000 to 500 cm^−1^.

#### 2.3.2. Thermogravimetric (TG) Analysis

TG analysis was carried out using an STA 449C (Netzsch Corporation, Selb, Germany) equipped with a thermal analysis data station at a heating rate of 10 K·min^−1^ under an Ar atmosphere. The sample was heated from 25 to 600 °C.

#### 2.3.3. Scanning Electron Microscopy (SEM)

SEM was conducted using a JEOL JSM 5900 LV scanning electronic microscope (SEM, Jeol, Tokyo, Japan) with a tension of acceleration of 20 kV to characterize the fracture surfaces of specimens.

#### 2.3.4. Mechanical Analysis

The tensile properties and compressive strength of the cured materials were recorded using a microcomputer control 858 Mini Bionix universal testing machine (MTS Systems (China) Co., Ltd., Shenzhen, China). The flexural and compressive properties were tested according to Chinese Standard GB/T 2567-2008. The samples’ shapes and sizes were as shown in Figure 1. The samples for compression testing and impact properties were rectangular and the samples for tensile properties were dumbbell-shaped, with sizes of 20 mm × 20 mm × 10 mm and 200 mm × 20 mm × 4 mm, respectively.

#### 2.3.5. Corrosion Resistance Test

The cured epoxy resin was soaked in either water, sodium chloride solution (10 wt %), sodium hydroxide solution (10 wt %), or sulfuric acid solution (10 wt %) for 100 h. Then, the compressive strength and tensile strength were tested.

## 3. Results

### 3.1. Mechanical Properties

Shown in Figure 2 are the stress–strain curves of the modified epoxy resins synthesized with different contents of prepolymer. According to the stress–strain curves, the modified epoxy resin exhibits two stages of deformation—elastic deformation and plastic deformation—during the tensile process. During the process of deformation, the respective tensile strengths of the modified epoxy resins were 13.1, 40.6, and 14.7 MPa. It is worth noting that the tensile strength was invariably higher than those in other results by 14–36% [30,31]. The respective maximum elongations at break of the specimens were 33.3%, 31.5%, and 28%. These results are consistent with the results of Bajpai, but the elongation at break in our results is 70% higher than that in his [32]. The tensile strength and the elongation at break of the modified specimens are obviously higher than those of the pure epoxy resin specimen. This may be caused by the flexible chain segments of the polymer polyols added into the chain of the modified EP and the interpenetrating network structure formed through the trimer [33]. The tensile strength of the 10% EP-PUP specimen is higher than those of the other two; this may be due to the fact that the epoxy resin in the 10% EP-PUP specimen is residual while the polyurethane prepolymer in the 30% EP-PUP specimen is residual, which leads to an incomplete graft reaction and an incomplete network structure.

The effect of the amount of prepolymer on the compressive and impact properties of the specimens is shown in Figure 3. It can be seen that the impact property first increased and then decreased with increasing prepolymer content. When the amount of prepolymer added to the epoxy resin was 20%, the material showed the best state of compressive and impact resistance. The value of impact performance was 76.6 kJ/m^2^, which is better than the 9.25 kJ/m^2^ in Park’s study and the 17.63 kJ/m^2^ in Hai-an Xie’s study [34,35]. Compared with that reported in other people’s research, the performance of the pressure resistance is greatly improved [36]. The results are in agreement with those shown in Figure 2. This is also mainly due to the addition of the flexible segments of prepolymer grafted onto the epoxy resin chain. As a result, the prepolymer was interspersed in the epoxy resin system and intertwined with the epoxy chain to form an interpenetrating network structure, thus changing the brittleness of the epoxy resin [37]. When the material is subjected to stress, the structure of the network can disperse the stress and the flexible chain can help to maintain a higher elongation at break. The compressive properties were also increased, which may be due to the advantages of the rigidity of the isocyanurate six-member ring in the trimer and the structure of the network. When the amount of prepolymer is too large, some of the prepolymer fails to produce a timely graft reaction; this results in self-polymerization, and the polymerization of the prepolymer exists in the form of small, isolated phases in the solidified compound. Furthermore, the compatibility between PUP and EP becomes worse, leading to a decline in mechanical properties. The results show that the modified epoxy resin has the best performance when the prepolymer content is 20%.

### 3.2. Scanning Electron Microscopy Analyses

The SEM images of the impact fracture surfaces of the cured products are shown in Figure 4. It can be observed in Figure 4a that the impact fracture surface of the pure EP is orderly, uniform, and smooth, showing a typical brittle fracture morphology [38]. This indicates that the pure epoxy resin only has low resistance to crack growth, and its ability to resist crack expansion is very weak. Thus, the crack grows rapidly, and brittle fracture occurs. In Figure 4b, it can be seen that the grain of the fracture surface expands along the direction of impact force. There are some chain segments with a thick end and a thin end, which may be the modified EP, and a smooth and uniform fracture surface can also be seen, which may be the pure EP. Brittle fracture and ductile fracture are also shown in the impact fracture surface. In Figure 4d, the stripping surface between two phases can be seen; this may be superfluous prepolymer PUP forming a small, isolated phase through self-polymerization. Thus, an interface is formed between PUP and the modified EP, giving poor compatibility. Meanwhile, in Figure 4c, uniform fibers and the typical flexible fracture surface can be seen, which means that a complete network was formed [39]. This may be the reason for the results shown in Figure 2 and Figure 3.

### 3.3. FT-IR Spectra of the EP-PUP and Cured Samples

The structure of the modified epoxy was confirmed by FT-IR, as shown in Figure 5. The IR spectrum of PUP (line A) shows that the characteristic groups of the polyurethane prepolymer are the stretching vibration peak of the N–H bond at 3348.74 cm^−1^, the isocyanate group at 2273.65 cm^−1^, and the stretching vibration peak of carbonyl at 1730.17 cm^−1^. The characteristic absorption peaks of E-44 (C) are at 3472.79 cm^−1^ and 914.51 cm^−1^, respectively corresponding to the hydroxyl group and the epoxy group. Comparing line B with line A, the infrared peak of EP-PUP disappears at 2273.65 cm^−1^, while the peak at 1730.17 cm^−1^ does not change. Compared with E-44, the modified epoxy resin has an obvious change: the absorption peak at 3472.79 cm^−1^ becomes weaker and wider. However, the characteristic absorption peak of epoxy resin at 914.51 cm^−1^ was retained. Therefore, it is clear that the change group is only the isocyanate group of the prepolymer and the hydroxyl group of E-44. This indicates that the synthesis of the modified epoxy resin was accomplished by reaction between the isocyanate group and the hydroxyl group. So, the mechanism of the modified epoxy resin is thus deduced from Scheme 1 and Scheme 2.

Benefiting from the presence of PPG, as a flexible group embedded into the structure of the modified EP-PUP, and the structure of the interpenetrating network formed through graft copolymerization, the mechanical properties of toughness, impact and compression resistance, and elongation at break all showed excellent improvement. This is in agreement with the results presented in Section 3.1 and Section 3.2.

### 3.4. Thermal Curing Behavior of Different Proportions of Cured Samples

Figure 6 shows the TG curves of cured neat EP and EP-PUP samples investigated under Ar atmospheres, and the corresponding data are listed in Table 2. The change in the general trend of the TG curve shows that the thermal decomposition mechanisms of E-44 epoxy resin and EP-PUP are basically the same. As Figure 6 shows, the thermal decomposition of solidified EP-PUP compounds is basically divided into three stages. The first stage is when the decomposition temperature ranges from 0 to 310 °C and is mainly the volatilization of the remaining oligomer and residual moisture in the modified epoxy resin. The second stage (310~480 °C) is mainly caused by the elimination reaction of the terminal groups (hydroxyl groups, amino groups) of the modified epoxy resin cured structure. In the third stage, the thermogravimetric loss at 480 °C is reduced obviously, which is caused by the degradation of the main chain of the epoxy resin.

Table 2 shows that the temperature of 50% mass loss of 20% EP-PUP is 406.6 °C, and that of pure epoxy is 403.7 °C. When the mass loss rate reaches the maximum, the corresponding temperature of 20% EP-PUP is 425.5 °C and that of pure epoxy resin is only 376.7 °C. It was found that the thermal stability of pure epoxy resin is not as good as that of 20% EP-PUP. The results show that the modified epoxy resin with a complete interpenetrating network has higher thermal stability. 

### 3.5. Corrosion Resistance Test

The corrosion resistance of the modified epoxy resin was studied by immersing the samples in either water, sodium chloride solution (10 wt %), sodium hydroxide solution (10 wt %), or sulfuric acid solution (10 wt %) for 100 h. As shown in Figure 7, the compressive strength and elongation at break of the specimens showed little change after being dipped into the corrosive liquid for 100 h. The compressive strength of the sample dipped in sulfuric acid solution decreased by 3~4 MPa. This decrease was within the reference range (−5% to +5%). The results show that the mechanical properties of the modified epoxy resin were not damaged after the corrosive process, and the modified epoxy resin has excellent corrosion resistance.

## 4. Conclusions

In summary, for the first time, a novel modified epoxy resin with high mechanical properties was prepared by a graft copolymerization method using TDIT. PPG, with its property of flexibility, was successfully embedded into the chains of PUP, and EP-PUP with an interpenetrating network structure was obtained. Due to the presence of the flexible PPG group and the complete structure of the interpenetrating network, the modified epoxy resins showed excellent mechanical properties including compressive strength, impact strength, tensile strength, and fracture elongation. With a mass ratio of PUP to EP close to 20%, the impact strength of the EP-PUP was significantly increased to 76.6 KJ/m^2^ and the fracture elongation reached 32.3%. The compressive strength and tensile strength were increased to 184.8 and 40.6 Mpa, respectively. Further, the modified epoxy resin also has excellent heat resistance and corrosion resistance. The present work provides a simple and convenient pathway to prepare a new modified epoxy resin with high performance which may have potential applications in a wide range of fields.

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
