# Peer review of "Preparation and Properties of Toluene-Diisocyanate-Trimer-Modified Epoxy Resin"

_polymers, 2019, doi:10.3390/polym11030416_

Round 1

Reviewer 1 Report

The manuscript "Preparation and properties of Toluene Diisocyanate Trimer Modified Epoxy Resin” regards the development of  novel modified epoxy resin with a structure of interpenetrating network 10 as grouting materials with high toughness was prepared by a method of graft copolymerization 11 between polyurethane prepolymer (PUP) trimer an epoxy resin(E-44).  

The study is relevant, the paper is well organized and the topic is up to date.

I recommend this work for publication.

Few points should be addressed before paper can appear in the journal, in fact the authors should compare their results with those obtained in the literature; in particular in the works in which an increase in the toughness of the epoxy resin, by different modifications of the matrix, is obtained, highlighting, if possible, the advantages of the modification described in  own manuscript. For instance with the  work described in DOI: 10.1007/BF01174528; DOI: https://doi.org/10.1016/j.compositesb.2017.07.021; DOI: 10.3390/ma10101131

Author Response

Response to Reviewer 1 Comments

Point 1: Few points should be addressed before paper can appear in the journal, in fact the authors should compare their results with those obtained in the literature; in particular in the works in which an increase in the toughness of the epoxy resin, by different modifications of the matrix, is obtained, highlighting, if possible, the advantages of the modification described in  own manuscript.

Response 1: Line 106-110  During the process of deformation, the tensile strength of the modified epoxy resin is 13.1 MPa, 40.6 MPa, 14.7 MPa, respectively. It is worth to remark that the tensile strength could be invariantly higher than results of others by a value of 1436% [30,31]. And the maximum elongation at break of the specimen are 33.3%, 31.5% and 28%, respectively. The results are consistent with results of Bajpai, A. , but the elongation at break is 70 percent higher than his[32].

Line 123-126  The value of impact performance is 76.6 kJ/m2, which is better than 9.25 kJ/m2 in Park's study and 17.63 kJ/m2 in Hai-an Xie's study[34,35]. Compared with other people's research, the performance of pressure resistance has been greatly improved[36].

I have mainly added a comparative analysis of the results of other researchers in these two places, and in other places I have only inserted the relative literature.

Reviewer 2 Report

The article claims to demonstrate an advantage of combining flexible segments and the interpenetrating network structure what was clearly demonstrated and supported by analysis.

There are a number of misprints and small typos present in the paper:

Line 46: EP abbreviation used for the fisrt time

Line 66: ratio(… space missing

Line 68: an should be a

Line 73: cured

Line 81: -1 should be superscript

All figures has problems to be read in B/W – please fix it

Figure 7. Please add error bars to Figure 7. (Since you mentioned that measurements for H2SO4 were within the error range – do you mean instrumental error?)

Regarding discussion of the results:

This is not the first paper exploiting such an idea (of combining cross linking prepolymers with epoxy resin) and it will be nice to see references as a support for your discussion section. For instance for the parts Line Line 115-121 and Line 128-135.

Author Response

Response to Reviewer 2 Comments

Point 1:  Line 46: EP abbreviation used for the fisrt time

Response 1: Line 26  Epoxy resin (EP)

Point 2: Line 66: ratio(… space missing

Response 2: Line 64-65  ratio(in Table 1)

Point 3: Line 68: an should be a

Response 3: Line 66  a transparent liquid 

Point 4:Line 73: cured

Response 4: Line 71  and then cured naturally

Point 5:Line 81: -1 should be superscript

Response 5: Line 79  a heating rate of 10K•min-1 

Point 6:All figures has problems to be read in B/W – please fix it

Response 6: The following figure changes have been made:

Figure 2. Stress-strain curves of PU modified epoxy withdifferent raw material ratio

Figure 3. Impact toughness of modified epoxy resin

Figure 6. TG acurves for the different cured samples.

Figure 7. Mechanical properties of corroded samples at 0℃. (a) The compassive strength of modified epoxy under H2SO4, NaOH and NaCl solutions. (b) The tensile strength of modified epoxy under H2SO4, NaOH and NaCl solutions.

The other pictures do not affect the reader's reading, so they have not been modified.

Point 7:Figure 7. Please add error bars to Figure 7. (Since you mentioned that measurements for H2SO4 were within the error range – do you mean instrumental error?)

Response 7: Line 213 This decrease was within the reference range.(-5%~+5%)

I'm sorry it was an error in my expression.The properties of the product after soaking in sulfuric acid were within the range of reference values which allow the corrosion resistance to be reduced.

Figure 7. Mechanical properties of corroded samples at 0℃. (a) The compassive strength of modified epoxy under H2SO4, NaOH and NaCl solutions. (b) The tensile strength of modified epoxy under H2SO4, NaOH and NaCl solutions.

Point 8:This is not the first paper exploiting such an idea (of combining cross linking prepolymers with epoxy resin) and it will be nice to see references as a support for your discussion section. For instance for the parts Line Line 115-121 and Line 128-135.

Response 8: Line 106-110  During the process of deformation, the tensile strength of the modified epoxy resin is 13.1 MPa, 40.6 MPa, 14.7 MPa, respectively. It is worth to remark that the tensile strength could be invariantly higher than results of others by a value of 14–36% [30,31]. And the maximum elongation at break of the specimen are 33.3%, 31.5% and 28%, respectively. The results are consistent with results of Bajpai, A. , but the elongation at break is 70 percent higher than his[32]. 

Line 123-126  The value of impact performance is 76.6 kJ/m2, which is better than 9.25 kJ/m2 in Park's study and 17.63 kJ/m2 in Hai-an Xie's study[34,35]. Compared with other people's research, the performance of pressure resistance has been greatly improved[36].
